

# Glider Technology for Ocean Observations: A Review

David Meyer[*]

[*]Leibniz Institute for Baltic Sea Research Warnemünde, Seestrasse 15, Rostock, 18119, Germany

*Correspondence to:* David Meyer (david.meyer@io-warnemuende.de)

**Abstract.** The total number of drones used in the air, on the land and in the water is growing in recent years. This review focuses on ocean robots and in particular on glider technology which seems to be one of the most promising oceanographic tools for future marine research.

Glider are remotely controlled underwater vehicles carrying out missions with lifetime exceeding months, traveling thound-

sands of kilometers, even through harsh environments. Depending on their scientific payload, they can be used for very specific research tasks, as well as for common environmental monitoring purposes. In combination with other technologies (e.g. moorings, satellites, drifters, floats) and as a part of existing observation networks they are of great advantage and thus help to get a more synoptic view on the world's oceans.

This review covers a wide range of topics - from the history and the development of glider technology to its application in a

variety of field studies regarding marine sciences. It offers a comprehensive overview of both, the technical and the scientific dimension, facilitating a fast access to the world of glider.

## 1 Introduction

A glider is a buoyancy-driven autonomous underwater vehicle (AUV) and meanwhile a widely used modern oceanographic tool. Its development has a long and absorbing history, which will be briefly described in the following paragraphs.

It all began after the second world war in the 1950s. At this time little was known about deep ocean circulation and an idea came up to use neutrally buoyant floats to directly measure low-frequency large-scale sub-surface currents (Stommel, 1955). The idea was simple but powerful. The float was made of an aluminium tube which is less compressible than seawater and therefore gains buoyancy when it sinks. Thus, depending on the ballasting process a float could be used to follow a constant pressure plane. The currents at that depth could then be obtained by tracking the float acoustically and integrate its drift over

the respective period. The first "Swallow"-floats (3 m, 10 kg) were deployed in 1955 over the Iberian Abyssal Plain by John Swallow himself (Swallow, 1955). Tracking of the floats were realized using hydrophones mounted on a ship and a 10 kHz acoustic signal to allow short-range navigation. The method was successfully demonstrated and the obtained data even accurate enough to show evidence for tidal variations. Afterwards floats were used more and more systematically to get a deeper knowledge about ocean circulation (e.g. Swallow and Hamon, 1959) and till the 1980s, several neutrally buoyant floats have

been developed and deployed. The life-time of such floats increased from several days to several weeks, and further to several months. Since ship-time is always a limiting factor and also cost-intensive, SOFAR-floats were developed by Tom Rossby



and David Webb and deployed in 1968 for the first time (Rossby and Webb, 1970). They could easily be tracked from several shore-based hydrophone sites in the western North Atlantic by using the SOFAR-channel of the water column for sound transmission (MODE Group, 1978). As a consequence, tracking of the float signals was possible at ranges of up to 1000 km and, hence, research vessels only had to be used for deployment and recovery. The resulting restriction of the SOFAR-floats to

depths close to the sound channel, however, finally leads to the development of a ship-based system for float communication whereby a signal transmitter was used, which simply could been attached on the CTD/water sampler package. The ability to interrogate from a wide range of depths ultimately allowed up to 18 floats at various depths to be tracked simultaneously and furthermore to carry out hydrographic observations at the same time (Swallow et al., 1974). A few years later, the development of moored hydrophones, so-called Autonomous Listening Stations, finally allowed tracking SOFAR-floats anywhere in the

oceans (Richardson et al., 1981). The problem with these floats was that they were relatively large and heavy (length: 5 m, weight: 430 Kg) and therefore expensive to deploy and to use. Reversing the direction of acoustic signalling to emitting moored systems and receiving floats ultimately led to the development of RAFOS floats (length: 2 m, weight: 10 Kg) (Rossby et al., 1986), which were much more cost-effective and easier to handle.

The application of floats during this time were numerous and significantly improved our understanding of oceanic eddy fields

and basin-scale mean circulation (Gould, 2005). The impact of the results provided by the float technology on marine science in general cannot be assessed high enough. The results revealed that the ocean is a highly turbulent system and ultimately lead to the discovery of the ocean mesoscale. And the success story of floats should be continued!

Another technical progress started the next step of float evolution. The use of satellites equipped with radar altimeters and

scatterometers in 1978 (SeaSat) allowed for the very first time to gather information about ocean circulation processes (sea level, surface temperature) on a global scale and thus emphasized the need to use in situ observing systems such as the float technology also on a global scale. Therefore an enhanced infrastructure and a higher level of coordination was required and for this purpose another big step was done by developing the Autonomous Lagrangian Circulation Explorer (ALACE) (Davis et al., 1992). The float was able to change its buoyancy by pumping a fluid from an internal reservoir to an external bladder, and,

thus it was possible for ALACE to surface and return to a pre-specified depth repeatedly. So far floats only had the capability to surface at the end of their mission by dropping a ballast weight. However, arrived at the surface, ALACE floats could now be tracked globally by ARGOS satellites using a 401 MHz signal. Profiling ALACE floats (P-ALACE) equipped with a CTD probe could even measure conductivity, temperature and pressure. ALACE, P-ALACE and SOFAR, but also other types of floats such as RAFOS (Rossby et al., 1994) and MARVOR (Ollitrault et al., 1994), were extensively used in the World Ocean

Circulation Experiment (WOCE) between 1990 and 1998 (WOCE IPO, 2002), but also in other studies. They were used, for instance, for the exploration of the intermediate water circulation in the Brazil basin (Boebel et al., 1999), the diapycnal mixing in the North Atlantic (e.g. Sundermeyer and Price, 1998) and during deep convection experiments in the Labrador sea (e.g. Lavender et al., 2002). In the last twenty years many different floats have been developed and produced (e.g. PROVOR & ARVOR (NKE-Instrumentation, Ifremer), APEX (Teledyne Webb), SOLO (Scripps), S2A (MRV Systems)).

Today, over 3000 of these floats are in the water to monitor the ocean on a global scale for temperature, salinity and velocity



data and the number of floats equipped with biogeochemical sensors for example for $O_2$ and $NO_3$ measurements is also grow-ing. This ambitious long-term project, known as ARGO (Array for Real-time Geostrophic Oceanography), started in the year 2000 and is actually one of the major components of the current ocean observing system (Roemmich et al., 2009). This clearly shows the development of float technology from an originally exclusive research tool to an oceanographic tool which addresses

issues of global socio-economic relevance such as climate change and sea-level rise.

The only problem which still remains by using a float is that its position cannot be controlled because it is always drifting with the currents. This problem was remedied by the glider concept, which was initially conceived by Henry Stommel (1989). In an Science Fiction article published in Oceanography he described a fleet of floats that „migrate vertically through the ocean

by changing ballast" whereby „they can be steered horizontally by gliding on wings at about a 35 degree angle". This article was visionary and provided the blue print for the future of glider technology. In the mid-1990s three programs supported by the Office of Naval Research (ONR) were created to develop several operational gliders: (1) Slocum Electric manufactured by Webb Research Corp (Webb et al., 2001), (2) Seaglider (Eriksen et al., 2001) built at the University of Washington and (3) Spray (Sherman et al. 2001) built at Scripps Institution of Oceanography (Fig. 1).

This article makes no claim to completeness but is instead intended to describe the main functional systems of these vehicles and to provide a comprehensive review of the studies that have been done to date with the participation of glider technology. The first part of this review gives a general description of the main glider components (e.g. buoyancy engine, hull design, navigation, communication, energy storage, and sensing capabilities) and thus helps the reader to get a brief overview of the

extensive literature. The second part focuses on the scientific results and touches the question of future perspectives. In this respect the limitations and the potentials of this relative new technology will be investigated. For other reviews of glider, see Davis et al. (2002), Rudnick et al. (2004), Griffiths et al. (2007), Johnson et al. (2009), Testor et al. (2010), Wood and Mierzwa (2013), and Rudnick (2016).

## 2   Design and construction

A glider is a small but powerful oceanographic tool. It can be used for transects, time series or roving assistance for research cruises. Typical values for glider length, diameter and weight are 180 cm, 30 cm and 50 kg. In this section glider design and construction will be investigated. In order to present a comprehensive overview, different technical parts of a gliding vehicle such as engine, communication, performance and navigation will be presented in the following paragraphs. Finally, other

important aspects such as sensor payload and the procedure of ballasting will be discussed.



## 2.1 Engine

The overall principle of glider engine concept is to change the glider volume and therefore to change glider buoyancy. The principle of volume change is either realized by pumping a low viscosity hydraulic oil back and forth between an internal and an external bladder or by pushing seawater in or out of a cavity, and was first used by ALACE floats (Davis et al., 1992). Thus,

the mass of the vehicle remains constant but its density changes relative to the surrounding water and thereby it rises or sinks. Unlike floats, glider cycle up and down through the water column in a saw tooth pattern, because their wings and their body lift are converting the vertical motion also into horizontal motion, both as the vehicle ascends and descends (Fig. 2). The energy for volume change can be provided from different sources. Seaglider, Spray and Slocum Electric have internal energy sources such as lithium or alkaline battery packages and thus can be classified as electrically driven vehicles.

Slocum thermal, as an exception, is a thermally driven glider which harvests the energy needed for the volume change from the oceans temperature gradient. The volume change in Slocum Thermal is associated with melting and freezing of a paraffin wax. In the warm surface water the wax is melting and in the cold deep water it is freezing. Melting and freezing of the wax results in expanding and contracting processes respectively and these processes are finally driving the transfer of a fluid between an internal and an external reservoir which is needed for the volume change. Due to this innovative concept thermally powered

vehicles have a range 3 to 4 times that of a similar electric-powered glider, always on the condition of existing temperature differences sufficient to generate the required energy to keep the thermal system running (Jones et al., 2007). In the Arctic and Southern Ocean, for instance, the application of thermally driven gliders is therefore limited. Furthermore, Slocum thermal is still in development phase and is not commercially available yet. Nevertheless, it is a powerful propulsion concept and certainly will be used extensively in the future for glider as well as for float technology. A greater flight efficiency and expanded mission

capability is allowed by a hybrid engine as it is used in Slocum electric. The buoyancy of this special type of glider is adjusted by using a piston to flood or to evacuate a compartment with seawater instead of moving oil in or out of an external bladder. Furthermore, Slocum electric is blending both thruster propulsion and buoyancy engine and therefore is the only glider that is able to generate horizontal motion from internal energy. This allows, for instance, to break through a strong pycnocline, overcome stronger current regimes or to fly horizontally along the shelf until the open ocean is reached (Jones, 2012).

In contrast to Seaglider and Spray, which are using high-pressure wobble-plate reciprocating (multi-stroke) hydraulic pumps, Slocum glider use a more robust single-stroke pump to drive their buoyancy engine. Due to the high power of large-volume single-stroke pumps Slocum glider have a rapid maneuverability and therefore are very well suited for shallow water operations (30 m - 200 m water depth). The Slocum buoyancy engine, however, is also available in a deep configuartion (1000 m water depth). Maximum depth of operation for Seaglider and Spray are 1000 m and 1500 m respectively.

In summary, it can be stated that in contrast to traditional propulsion-driven autonomous underwater vehicles (AUVs), glider change their volume and therefore their buoyancy to generate forward motion. This technique is responsible for relatively slow glide speeds (20 - 30 cm/s) but is highly energy-efficient. Thus glider, which are in fact low-power AUVs, can be deployed for a long time and for a great distance with minimal dependence on large research vessels. Therefore, glider are more suitable for



sustained ocean observation than other AUVs.

## 2.2 Hull

The hull of a glider is another important part. It is needed to provide a streamlined hydrodynamic shape, to resist external
pressure and therefore to protect the electronics. Furthermore it is needed to determine drag and compressibility which have
an impact on important operational parameter such as duration and, consequently, cost of a deployment, especially for electric
powered gliders.

The hull construction differs from glider to glider. Seaglider has the most complex shell employing a fiberglass fairing which
encloses an aluminium hull with neutral compressibility similar to that of seawater. This is realized by deflecting arched panels
making up the inner hull. Compared with a stiffer hull, that kind of shell construction can save pumping energy and therefore
allows a greater operational depth. Any trapped air between the external and internal hull is vented before each dive by ports
which are integrated in the fairing. In contrast to that, the hull of Slocum electric is a modular construction and consists of
composite carbon fiber fore and aft hulls and an aluminium payload bay in the middle to allow integration of ported windows
and sensor feeds (Jones et al., 2007). Slocum Thermal, relying on the energy available with thermal power, uses a shape that
simplifies construction and maximizes packing efficiency. The case of the Spray is built in three segments which are machined
from 6061 aluminium alloy in which stiffening hoops have been integrated to prevent buckling failure under pressure. It has a
finer entry shape than the Slocum electric and therefore is more energy efficient (50% less drag) (Sherman et al., 2001). Thus
Spray, for instance, is very well suited for long-range, long-duration and deep ocean use.

## 2.3 Navigation

To navigate from one waypoint to the next a glider has to follow a certain track underwater. This targeted underwater move-
ment, called flying, is characterized by two parameters. The glide angle and the heading. To be able to adjust these parameters
the live performance of the glider has to be monitored constantly. Therefore glider navigation includes both monitoring perfor-
mance and subsequent adjustment of glide angle and heading.

Monitoring is performed by a state-of-the-art TCM2 Electronic Compass Sensor Module combining a 3-axis magnetometer
and a 2-axis tilt sensor. Adjustment of glide angel and heading is controlled by pitch and roll, respectively. This in turn is
achieved by axial translation (Spray, Slocum, Seaglider) and rotation (Spray, Seaglider) of internal battery packs, which results
in typically turning radii of 20 m - 30 m. Slocum glider, however, utilizes a tail fin rudder to change heading and thus have the
tightest turning radius (approximately 7 m). Therefore, Slocum glider are very well suited for shallow water operations and
littoral environments (Davis et al., 2002). Although Spray and Seaglider using internal mass rotation instead of a rudder to roll,
their turning dynamics are the opposite of each other. Due to the different wing position (Seagliders wings are aft and Spray
gliders wings are centred) for the Seaglider a roll to the left produces a left turn but for the Spray a roll to the left produces a
right turn.



However, further differences exist with regard to underwater navigation. A very serious challenge is to stay away from the seafloor during a mission to avoid damage or the loss of the vehicle. In order to prevent "'bottom sampling"' glider constantly measure altitude and as a result change heading and glide angle. Slocum and Spray measure altitude using an acoustic altimeter. In contrast to that, Seaglider determines altitude from glider depth and bathymetric data, which is digitally stored on-board.

Anyhow, due to possible malfunctions of the altitude sensor Slocum and Spray user must have bathymetry data in a mission area and monitor glider depth to ensure that the glider is not colliding with the seafloor.

However, another important parameter for glider navigation, in particular for adjusting their flight path during their trajectory, is the speed and the direction of ambient currents. To take this effect into account, all three gliders described here use GPS navigation at the surface to dead-reckon toward commanded waypoints. Depth-averaged horizontal velocities are then deter-

mined over each dive cycle by using the differences between dead-reckoned and actual displacements. On the other hand ocean currents such as tides can be utilized to navigate to a certain location and thus to save pumping energy. This can be done, for instance, by deploying the glider on the seafloor during unfavourable current directions (e.g. ebb) and by retrieving it from the seafloor when the current direction has changed (e.g. flood). At this point, it should be noted that this kind of operation could endanger the whole mission as well as the glider itself unless users have complete knowledge of 3D tidal currents in the

mission area.

## 2.4   Communication

Glider frequently come to the sea surface to communicate with mission control by using the Iridium satellite phone system. Antennas can be housed in a tail fin (Slocum), in a tail string (Seaglider) or in a wing (Spray). They are raised above the surface

while the vehicle is communicating or obtaining a GPS fix. In high sea-states, however, loss of performance of communication systems can occur, but glider typically have enough internal memory for message buffering and data storage. Some vehicles such as Slocum glider use an external airbladder which can be inflated at the surface by a small pump to increase surface buoyancy and pitch moment to achieve a higher antenna position. This system in turn can be used for emergency case scenarios when the buoyancy engine fails and the glider is not able to surface. Furthermore, Slocum and Spray are using ARGOS

as a backup location and telemetry system. A Freewave 900 MHz modem master for line of sight radio communications is accomplished only by Slocum.

Frequent surfacing of the gliders and the used worldwide communication system allow permanent adjustment of flight parameters but also optimisation for features like thermo- and haloclines. Even for under ice trajectories communication solutions have been developed. Acoustic receiver integrated into glider can be used to supplant the unavailable GPS-signal and ultimately to

facilitate long range navigation in ice-covered environments (Jones, 2012). Seagliders, for instance, were successfully deployed across Davis Strait operating fully autonomous under ice for several weeks (Lee et al., 2010). Furthermore, Song et al. (2014) have recently demonstrated a feasibility of using glider as a mobile communication gateway.

For land- and ship-based glider communication usually a linux-based communication center is used which is provided by the individual company. For Slocum it is called Dockserver, for Seaglider Basestation and for Spray groundstation. Basically, it is





a computer set up to communicate remotely via internet and satellite link to a glider or a glider fleet. This shore-side computer end can work from any internet hub in the world and the current glider mission can even be tracked by multiple terminals simultaneously. Thus, glider missions can be collaboratively piloted by different groups working in different locations. Data transmission needs normally 10 to 30 minutes and includes information on positioning, scientific records and other perfor-
mance indicators (Ruiz et al., 2012a). Oceanographic and engineering data are then available for web based presentation. After mission evaluation, path planning algorithms can be used for changing important flight parameters and finding an appropriate glider setting (Smith et al., 2011).

## 2.5    Energy

The amount of energy carried by a glider affects how long the glider can stay out in the water before it must be recovered and therefore it determines mission length and duration.

Since the major part of the available energy is consumed by the buoyancy pump (~70 %), the depth of the dives and the stratification of the ocean are critical parameters. This means glider batteries last longest when dives are deep, slow, and occur in water that has little change in density. Despite of this, a few weeks of glider application in shallow waters still provides the
user thousands of vertical casts. Ultimately, energy budget calculations should be carried out in advance to ensure success of the particular deployment.

Stored energy for the buoyancy engine and the other consumers in electric gliders (processor, telemetry, scientific payload) come from primary lithium batteries. On the one side, lithium batteries have a high energy density and thus they can be used for multiple deployments with reduced mission preparation time. On the other side, they are considered a hazardous material
and thus transport and handling of gliders using that kind of batteries is more difficult. Furthermore, lithium batteries are non-rechargeable. Slocum, however, can also be fitted with alkaline batteries, which are safer and less expensive, but then mission length is reduced.

In summary, available current battery packages are satisfactory and enable basic long-term deployments up to several months but a continuing improvement in high capacity power storage systems is still desirable.

## 2.6    Modes of operation

Depending on the type of examination and mission requirements a glider can be used in different modes. The two most important modes are the survey and the virtual mooring mode.

During the survey mode the glider is flying along a track of waypoints. This can be a repeat section or a long distance transit. To get access to the mesoscale variability usually a relatively short section is chosen and sampled repeatedly. A long survey
with a solitary glider, however, can also be of great importance. The crossing of the Atlantic by RU 27 ("Scarlet Knight") in 2009 draw a lot of attention to glider technology and therefore was important for bringing this new oceanographic tool into the public mind.

During the virtual mooring mode the glider stays at one single target location as long as ambient current velocities are not ex-





ceeding glider speed (Hodges and Frantantoni, 2009; Alenius et al., 2014; Karstensen et al., 2014). Thus, by holding horizontal position nearly constant, a glider is measuring continuous depth profiles while ascending and descending through the water column. In contrast to a profiling mooring, which also can be used for measuring continuous depth profiles, a glider exhibits a better cost-benefit ratio, mainly due to its small size and its independence of large vessel operations.

Finally, a combination of both modes might be the most promising sampling strategy. Thus, a glider can be deployed close to the shore by a small research vessel, transit to a predetermined location using the survey mode, virtually moor itself for a while, and return to the shore by using the survey mode again for recovery. Short repeated sections, as mentioned above, also are of great interest, but suffer from low glide speeds ($\sim$ 20 km/d). Spatial structure is often changing on short time scales (days to weeks), and therefore glider data usually is aliased by temporal variability. Ultimately, the use of multiple vehicles

and a mixture of sampling modes might be the best way to study phenomena on a wide variety of time and space scales. The deployment of a glider fleet already was successful tested in the field and is still a focus of ongoing research.

Although survey and virtual mooring mode are the most relevant modes, also other modes such as surface, sub-surface and drift mode can be used for particular purposes. Surface mode, for instance, is used for positioning the antenna for data telemetry, and is important for monitoring live performance of the glider. On the contrary, sub-surface mode is needed for performing

multiple dive cycles before surfacing. On the one side, this can be helpful to avoid damage or the loss of the vehicle in a high traffic area, on the other side, it can reduce the time which is spent at the surface and thus improve temporal resolution of the gathered data. Drift mode maintains neutral buoyancy at any depth and can be used to transform a glider into a float or to park it on the seafloor.

The potential applications and possibilities of mode combination are various and only limited by the energy which can be

stored by the glider. Maybe someday in a more advanced world a glider will not only move through the water, but also be able to return back to mission control by flying through the air. An air-ocean robot could help to reduce deployment and recovery costs to a minimum and concurrently provide a more holistic view of the earth system.

## 2.7 Ballasting

Gliders are complex technical platforms and the preparation of a glider for a particular mission is not trivial. Before it can

be deployed in the field a ballasting process has to be carried out in the lab to ensure a proper functioning of the glider. If the vehicle is too light maximum operational depth cannot be reached and, if is too heavy it is likely that surfacing will be impossible.

Ballasting is a procedure for adding or subtracting weights to trim the glider for a given range of water densities. The initial ballasting is done in a test tank filled up with seawater of a certain temperature and conductivity. The temperature and conduc-

tivity of the tank water is then measured by the glider's CTD and a software program is finally used to calculate the weight which must be added or subtracted to the glider to reach optimal buoyancy for target water properties.

To trim flight parameters such as pitch and roll centers a test ballasting is carried out during a few initial shallow dives at the deployment area. After test diving is completed the glider is programmed to dive to its maximum depth and the real mission can be started. Due to the permanent live monitoring of the glider's performance necessary post-adjustments can be made during



the mission while the glider is at the surface by using the Iridium satellite phone system. Post-adjustments are necessary, for instance, if the weight and thus the ballasting of the glider changes due to environmental conditions. This can be the case, for example, in extreme environments such as polar or sub-tropical and tropical waters. In polar waters, freezing of wind blown ocean spray might be a problem with regard to the glider performance. In sub-tropical and tropical waters, biofouling has to

be taken into account (Lobe et al., 2010). Especially, if the glider is a shallow type glider operating in the upper euphotic layer of the ocean, where sunlight causes growth of mussels and algae on the gliders hull. In both cases there is a threat of losing the capability of surfacing due to the additional weight. Ultimately, glider operators need to have high-level knowledge in ballasting, but also in programming, communications, engineering, and path planning to minimize risks which could lead to the abort of a mission.

## 2.8  Scientific payload

The glider itself with all its communication and navigation technology and buoyancy engine is ultimately incomplete and not valuable without its scientific sensors measuring oceanographic parameters.

Each glider can carry a certain number of such sensors, which either can be mounted in a modular center payload bay (Slocum),

directly on the hull (Seaglider, Spray) or aft of the hull in a flooded compartment (Slocum, Seaglider, Spray). Currently commercial available sensors are various including CTDs (free flow, pumped), dissolved oxygen sensors, sensors for backscatter and fluorescence, PAR sensors (Photosynthetically Available Radiation), echo sounder, PAMs for passive acoustic monitoring, ADCPs/DVLs for current measurements, and sensors for nitrate and turbulence. Sensors for pH, $CO_2$ and a fluorometer also will be available in the near future.

Sensor calibration is carried out by the glider company. This means that the vehicle must be transported to the calibration lab of the company which takes a lot of time and money. Furthermore, cross-calibration with other ship measurements is recommended to take the drift of the sensors into account which is naturally occurring during long term deployments.

The number and the type of the used sensors is depending on the main goal of the mission. For monitoring bio-mass, for instance, an echo sounder must be installed, whereas backscatter and turbidity are important for oil exploration. For environ-

mental monitoring chemical, optical and physical sensors are needed. For monitoring man-made noise, supporting seismic operations or supporting anti-submarine warefare PAMs are more crucial. As a consequence, the more sensors are used, the more informations can be extracted. The tendency is, therefore, not to deploy a single glider, but a fleet of gliders equipped with different sensor suites. Unfortunately deploying and coordinating a whole fleet of gliders is relatively expensive and very complex in the implementation with regard to path planning.

Finally, a lot of different sensors for a broad range of applications already exist (Rudnick, 2016). However, for increasing the spectrum of science sensors steady development is still necessary. The future of in situ sensors has already been mapped out. Sensors need to be robust, low cost and hydrodynamic. Furthermore the most difficult challenge is to minimize their size and power consumption, since the size of scientific payloads for gliders is limited as well as the energy which the vehicle can carry.





By considering this requirements integrating new sensors will be successful and thus help to open up new possibilities to get a more holistic biogeochemical picture of the oceans.

## 3  Applications

Glider can be deployed in aquatic environments that are hostile and hard to reach. Furthermore they enable sampling at relevant temporal and spatial scales while providing a continuous observing presence to capture episodic events and characterize longer-term phenomena (Dickey et al., 2008). Tab. 1 shows a summary of relevant studies published in peer reviewed journals since 2008. There are five groups which can be derived from this literature compilation.

The first group was dealing with the observation of coastal, meso- and submesoscale dynamics. Main subjects of interest of these studies were: eddies (Hatun et al., 2007; Martin et al. 2009; Todd et al., 2009; Baird et al., 2011; Baird and Ridgway, 2012; Pelland et al., 2013; Thompson et al., 2014; Thomsen et al., 2015; Karstensen et al., 2016; Schütte et al., 2016), fronts (Ruiz et al., 2009; McClatchie et al., 2012; Cenedese et al., 2013; Timmermans and Winsor, 2013; Todd et al., 2013; Heywood et al., 2014; Mazzini et al., 2014; Pietri et al., 2014; Piterbarg et al., 2014; Powell and Ohman, 2015a), and upwelling (Davis, 2010; Adams et al., 2013; Pietri et al., 2013; Karstensen et al., 2014; Schaeffer and Roughan 2015), but also Rossby waves (Nicholson et al., 2008; Webber et al., 2014), river plumes (Castelao et al., 2008a; Castelao et al., 2008b), oxygen minimum zones (Pierce et al., 2012; Pizarro et al., 2015), mesoscale dynamics caused by typhoons (Mrvaljevic et al., 2013), the upper ocean response to hurricanes (Domingues et al., 2015), and the physical-biogeochemical coupling related to phytoplankton blooms (Perry et al., 2008; Frajka-Williams et al., 2009; Hodges and Frantantoni, 2009; Xu et al., 2011; Mahadevan et al., 2012; Swart et al., 2015; Thomalla et al., 2015) were investigated.

Glider provide high-resolution data in three dimensions, contributing to the characterization of mesocale (Pascual et al., 2013) and submesoscale processes (Niewiadomska et al., 2008). Although glider velocities are likely biased due to the low forward speed, they can help to obtain a detailed and quantitative representation of these features (Fan et al., 2013). In particular, the use of gliders in combination with altimetry is very promising and certainly will be used extensively in the near future to revise the general pattern and the associated forcing of ocean circulation (Bouffard et al., 2010). Aliasing effects can be reduced by enhancing the Nyquist wave number and frequency, for instance, by reducing the sampling depth (Rudnick and Cole, 2011) or by using a glider fleet instead of using only a single vehicle.

The second group was dealing with mixing processes and the transport of water and energy, whereby gliders were used mainly for measurements of velocities and turbulence (Merckelbach et al., 2010; Frajka-Williams et al., 2011; Fer et al., 2014; Peterson and Fer, 2014; Farrar et al., 2015; Palmer et al., 2015). Most of these studies were interested in Boundary Current Systems such as the California Current (Davis et al., 2008; Todd et al., 2011a; Todd et al., 2011b; Todd et al., 2012; Johnston and Rudnick, 2015), the Kuroshio Current (Gawarkiewicz et al. 2011; Rudnick et al., 2011; Lien et al., 2014), the New Guinea Coastal Undercurrent (Davis et al., 2012), the Norwegian Atlantic Current (Høydalsvik et al., 2013), the Northern Equatorial





Current (Schönau and Rudnick, 2015), the South Equatorial Current (Gourdeau et al., 2008), the Loop Current in the Gulf of Mexico (Brickley et al., 2012; Todd et al., 2015), or the East Australian Current (Boettger et al., 2015; Schaeffer et al., 2016). Other important fields of application were the Western Mediterranean, the Tasman Sea, the Faroe Bank channel and the South China Sea. The focus of these studies was mainly on cyclonic circulation (Heslop et al., 2012), winter intermediate water

formation (Juza et al., 2013), vertical mixing (Ruiz et al., 2012b), sub-surface geostrophic currents (Alvarez et al., 2013), water mass interaction (Beaird et al., 2012; Beaird et al. 2013; Ullgren et al., 2014), internal tides generated at submarine ridges (Johnston et al., 2015) and internal waves generated by barotropic tidal currents (Johnston et al., 2013; Rainville et al., 2013; Rudnick et al., 2013; Alford et al., 2015).

All of these studies have shown that with the unique spatial and temporal coverage provided by the gliders the mean and the

variability of currents as well as the characteristics of highly dynamic and energetic processes such as internal waves can be estimated with a high accuracy.

Group three used glider data to improve or to validate ocean models. Models used for this experiments were, for instance, the Navy Coastal Ocean Model (NCOM), applied for the Monterey Bay area (Chao et al., 2008; Chao et al., 2009; Ramp et al.,

2009; Shulman et al., 2009; Pan et al., 2011), the HYbrid Coordinate Ocean Model (HYCOM), applied for the Mid-Atlantic Bight (Wilkin and Hunter, 2013; Xu et al., 2013), the Regional Ocean Modeling System (ROMS), applied for the New York Bight (Zhang et al., 2010) and the southwest Pacific Ocean (Hristova et al., 2014), the Mediterranean Sea general circulation model, applied for the Eastern Mediterranean Sea (Dobricic et al., 2010), the French coastal operational forecasting system PREVIMER, applied for the Western Mediterranean Sea (Mourre and Alvarez 2012; Mourre and Chiggiato, 2014), the West

Florida Shelf coastal ocean model and the Massachusetts Institute of Technology general circulation model (MITgcm), applied for the Gulf of Mexico region (Gopalakrishnan et al., 2013; Pan et al., 2014; Rudnick et al., 2015), the Bluelink forecast system and the coastal ocean model of Tasmania, applied for the south-east Australian territory (Oke et al., 2009; Jones et al., 2012), and the Harvard Ocean Prediction System (HOPS), applied for the North-Atlantic (Gangopadhyay et al., 2013) and the California Current region (Haley et al., 2009).

In all studies assimilation of glider data improved model results significantly. For instance, Jacox et al. (2015) found that using glider data can help to considerably enhance estimates of primary production. Finally, ocean observing system simulations experiments clearly showed the high potential of glider data assimilation with regard to the correction of model simulations (Melet et al., 2012) and the efficiency of using a glider fleet instead of using a single glider to study a certain area (L'Heveder et al., 2013; Alvarez and Mourre, 2014).


Group four focused on the topics of acoustic detection of biological and geological activity. Glider were used, for example, for mapping an erupting submarine volcano in the south-western Pacific Ocean (Matsumoto et al., 2011), for detecting and allocating sounds in the Gulf of Mexico produced by fish (i.e. toadfish and red grouper) (Wall et al., 2012), or for estimating zooplankton biomass off the coast of California (Powell and Ohman, 2012; Powell and Ohman, 2015b) and in the Southern

Ocean (Guihen et al., 2014). Furthermore, mammals such as walrus and whale were studied by using gliders in the North



Atlantic, in the Pacific and in the Arctic Ocean (Baumgartner and Fratantoni, 2008; Klinck et al., 2012; Baumgartner et al., 2013; Baumgartner et al., 2014). Finally, some studies tried to link the hydrographic data to biological activity, without using acoustic measurements, investigating penguins along the Western Antarctic pensinsula (Oliver et al., 2013; Schofield et al., 2013) or seals off the Washington coast (Pelland et al., 2014).

Group five was dealing with sediment transport and resuspension occurring during a hurricane and tropical storms in the North Atlantic and in the Western Mediterranean (Glenn et al., 2008; Miles et al., 2013; Miles et al., 2015a; Bourrin et al., 2015). Glider observations helped to get a better understanding of the alongshore sediment transport on continental shelves and coincidently provided new challenges for coupled physical-sediment transport models.

However, also other studies were carried out taking advantage of glider technology, but which cannot be assigned to one of the five main research areas described above. Topics of these studies are various including, common environmental monitoring of hydrographic properties (Castelao et al., 2010), estimation of net community production (Alkire et al., 2012; Alkire et al., 2014) and aggregate flux (Briggs et al., 2011), determination of the ocean mixed layer depth (Chu and Fan, 2010), investigation of

mesophotic reef ecosystems at the West Florida Shelf (Dalgleish et al., 2012), cross-shelf transport, biogeochemical variability and phytoplankton blooms in the Southern Ocean (Asper et al., 2011; Kohut et al., 2013; Kaufman et al., 2014; Haskins and Schofield, 2015), under-ice activities using acoustic navigation (Webster et al., 2014), catalyzing undergraduate education in ocean sciences and technology (Glenn et al., 2011), biological production near icebergs (Biddle et al., 2015), convection in the Labrador Sea (Eriksen and Rhines, 2008; Frajka-Williams et al., 2014), and ice shelf outflow in the Amundsen Sea (Miles et

al., 2015b). In addition, glider were used to determine wavelength and amplitudes of sea surface waves (Alvarez, 2015) and subsurface chlorophyll maxima (Hemsley et al., 2015). Furthermore, it has to be noted that glider are also used extensively by oil and gas companies and military organizations, for example, to monitor pipelines or for anti-submarine warfare.

At this point it should be mentioned that glider are mostly not used as a standalone application. In many studies reported here, they were integrated into consistent observation networks, ultimately to complement other observational techniques such as

long-range high frequency radar, floats, drifters, moorings, towed underwater vehicles and satellite remote sensing (e.g. Lee et al., 2010; Schofield et al., 2010; Rudnick and Cole, 2011; Alvarez and Mourre, 2012a; Ohman et al., 2013; Zhao et al., 2013; Curry et al., 2014). Cole and Rudnick (2012), for example, observed the annual cycle of thermohaline fluctuations in the central subtropical North Pacific and found that gliders are advantageous compared with satellites, floats, and long-term stations because they can sample the same section repeatedly and provide a large number of profiles over a short period of

time. In particular, satellite measurements greatly benefit from glider data, due to the gliders independence of clouds and its ability to resolve subsurface features.

Moreover, many workers applied a network of several glider platforms, instead of using only a single vehicle, ultimately to get a more synoptic description of spatially extended marine areas and to better resolve rapid physical transitions and biological responses (e.g. Leonard et al., 2010; Alvarez and Mourre, 2012b). It was found that if a sufficient number of glider is used a

high level of confidence in obtaining data is achieved (Brito et al., 2014). Furthermore, problems encountered during glider



mission (e.g. leaks, communications interrupts) are less common now than in the initial years (Schofield et al., 2007). Finally, it can be stated that glider provide an effective way to conduct ocean science surveys and, for some missions, to a great extent reduce the costs associated with research vessels.

## 4 Conclusions

The collection of papers in this review covers a wide range of topics - from the history and the development of glider technology to its application in a variety of field studies regarding ocean sciences. The author offers a comprehensive overview of both, the technical and the scientific dimension, facilitating a fast access to the world of glider. This work provided descriptions

of the main functional systems of glider (e.g. buoyancy engine, hull design, navigation, communication, energy storage, and sensing capabilities) in order to give scientists who are not familiar with this oceanographic tool a short introduction to the most important glider components. Furthermore fundamental research is highlighted demonstrating versatile possibilities of applications.

It was shown that glider are remotely controlled ocean robots carrying out missions with lifetime of several months, traveling

thoundsands of kilometers, even through harsh environments. We have seen that glider, depending on their scientific payload, can be used for very specific research tasks, as well as for common environmental monitoring purposes. Furthermore, it was outlined that gliders are best used in combination with other technologies (e.g. moorings, satellites, drifters, floats) to ensure a more synergistic approach to ocean research. Especially, integration of glider in existing observation networks is of great advantage and always lead to a more synoptic view on the world's oceans.

However, apart from the achievements made by glider so far, some parts of glider technology described in this article are still under further development. The performance of glider definitively will be considerably enhanced in the near future, enabling more improved ocean observations, including deep ocean sampling, extended deployment periods and a broader range of accessible parameters. Glider have begun to assume a role in modern oceanography and certainly will help to overcome present limitations of the under-sampled ocean. In summary, judging by the various applications reported in this article, the future of

glider technology in ocean sciences looks bright.

*Author contributions.* D. Meyer did the literature research and drafted the manuscript.

*Acknowledgements.* This work was supported by funding of the government of the federal state of Mecklenburg-Western Pommerania (STB-AUTSTAT), in the north eastern part of Germany, for further developments of the IOW Baltic Monitoring and long term data program.





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

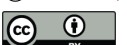

**FIGURES**

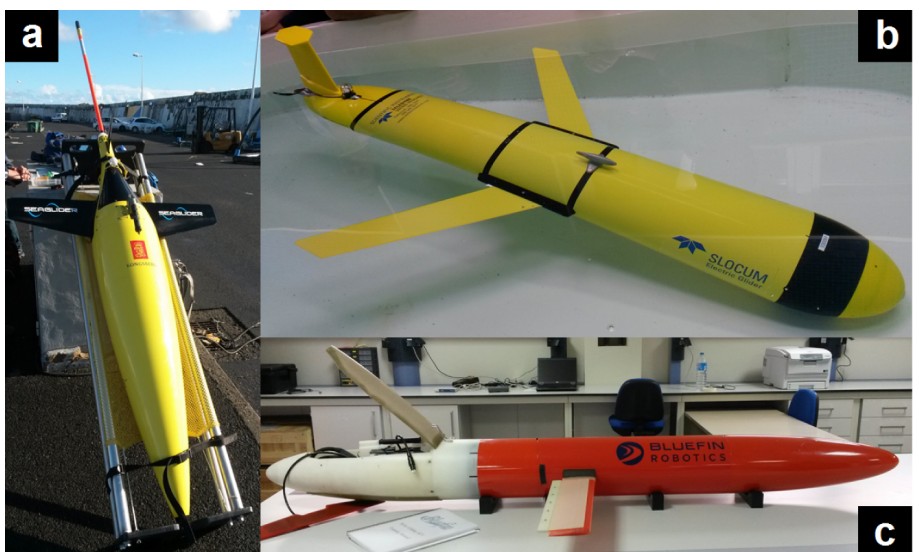

**Figure 1. a** Seaglider, **b** Slocum electric glider, and **c** Spray glider.

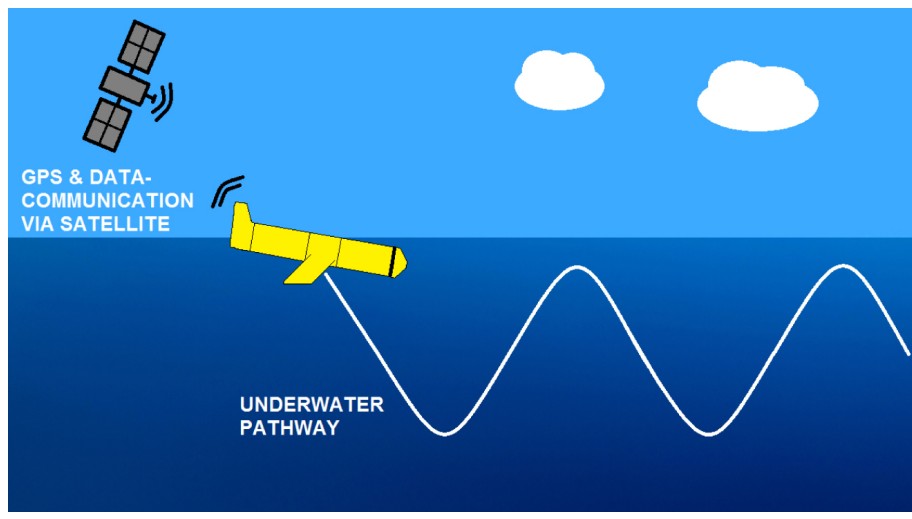

**Figure 2.** Sketch of a glider mission including flight path and surfacing process.




TABLES

**Table 1.** Overview of the 5 main application areas of glider technology in ocean sciences and corresponding relevant publications

| I. Observation of coastal, meso- and submesoscale dynamics (e.g. upwelling, eddies, rossby waves, fronts) | | | |
|---|---|---|---|
| Hatun et al., 2007 | Bouffard et al., 2010 | Mrvaljevic et al., 2013 | Thompson et al., 2014 |
| Castelao et al., 2008a | Davis, 2010 | Pascual et al., 2013 | Webber et al., 2014 |
| Castelao et al., 2008b | Baird et al., 2011 | Pelland et al., 2013 | Domingues et al., 2015 |
| Nicholson et al., 2008 | Rudnick & Cole, 2011 | Pietri et al., 2013 | Pizarro et al., 2015 |
| Niewiadomska et al., 2008 | Xu et al., 2011 | Timmermans & Winsor, 2013 | Powell & Ohman, 2015a |
| Perry et al., 2008 | Baird & Ridgway, 2012 | Todd et al., 2013 | Schaeffer & Roughan 2015 |
| Frajka-Williams et al., 2009 | Mahadevan et al., 2012 | Heywood et al., 2014 | Swart et al., 2015 |
| Hodges & Frantantoni, 2009 | McClatchie et al., 2012 | Karstensen et al., 2014 | Thomalla et al., 2015 |
| Martin et al. 2009 | Pierce et al., 2012 | Mazzini et al., 2014 | Thomsen et al., 2015 |
| Ruiz et al., 2009 | Adams et al., 2013 | Pietri et al., 2014 | Karstensen et al., 2016 |
| Todd et al., 2009 | Fan et al., 2013 | Piterbarg et al., 2014 | Schütte et al., 2016 |
| II. Mixing processes and transport of water and energy (e.g. variability of boundary current systems, internal waves) | | | |
| Davis et al., 2008 | Brickley et al., 2012 | Johnston et al., 2013 | Boettger et al., 2015 |
| Gourdeau et al., 2008 | Davis et al., 2012 | Juza et al., 2013 | Farrar et al., 2015 |
| Merckelbach et al., 2010 | Heslop et al., 2012 | Rainville et al., 2013 | Johnston et al., 2015 |
| Frajka-Williams et al., 2011 | Ruiz et al., 2012b | Rudnick et al., 2013 | Johnston & Rudnick, 2015 |
| Gawarkiewicz et al. 2011 | Todd et al., 2012 | Fer et al., 2014 | Palmer et al., 2015 |
| Rudnick et al., 2011 | Alvarez et al., 2013 | Lien et al., 2014 | Todd et al., 2015 |
| Todd et al., 2011a | Beaird et al. 2013 | Peterson & Fer, 2014 | Schönau & Rudnick, 2015 |
| Todd et al., 2011b | Cenedese et al., 2013 | Ullgren et al., 2014 | Schaeffer et al., 2016 |
| Beaird et al., 2012 | Høydalsvik et al., 2013 | Alford et al., 2015 | |
| III. Impact of glider data assimilation on ocean models | | | |
| Chao et al., 2008 | Dobricic et al., 2010 | Gangopadhyay et al., 2013 | Hristova et al., 2014 |
| Chao et al., 2009 | Zhang et al., 2010 | Gopalakrishnan et al., 2013 | Mourre & Chiggiato, 2014 |
| Haley et al., 2009 | Pan et al., 2011 | L'Heveder et al., 2013 | Pan et al., 2014 |
| Oke et al., 2009 | Jones et al., 2012 | Wilkin & Hunter, 2013 | Jacox et al., 2015 |
| Ramp et al., 2009 | Melet et al., 2012 | Xu et al., 2013 | Rudnick et al., 2015 |
| Shulman et al., 2009 | Mourre & Alvarez 2012 | Alvarez & Mourre, 2014 | |
| IV. Acoustic detection of biological and geological activity (mammals, fish, submarine volcanos, etc.) | | | |
| Baumgartner et al. 2008 | Powell & Ohman, 2012 | Oliver et al., 2013 | Guihen et al., 2014 |
| Matsumoto et al., 2011 | Wall et al., 2012 | Schofield et al., 2013 | Pelland et al., 2014 |
| Klinck et al., 2012 | Baumgartner et al., 2013 | Baumgartner et al., 2014 | Powell & Ohman, 2015b |
| V. Sediment transport/resuspension | | | |
| Glenn et al., 2008 | Miles et al., 2013 | Miles et al., 2015a | Bourrin et al., 2015 |