# Peer review of "Glider Technology for Ocean Observations: A Review"

_Ocean Science, 2016_

## Referee Comment (RC1) · Anonymous Referee #1 · 3 Jul 2016

This manuscript summarizes some historical and technological aspects of underwater glider technology. Moreover, the use of underwater glider technology in ocean research is analysed based on research topics covered by published papers.

All information that is presented has been published elsewhere (most of the papers are cited in the manuscript). I think the title is a bit misleading, although it claims to be "a review" the manuscript is not comprehensive - the author did not use the word "comprehensive" but given the existing, published reviews I would expect that yet another review would be more comprehensive or is topical e.g. glider use in ocean acoustics – both is not the case.

In fact it has been mentioned by the author (page 3, line 15) that nothing new is presented in the text and I am somewhat surprised that the manuscript has been sent out for review. . .

[Figure]

The only part that is a bit different from former published reviews is a summary on use-of-glider in ocean research, based on publications addressing certain research topics. However, a similar strategy has been published as a report some years ago (GROOM deliverable 2.01, see link below). Moreover, a recent paper in press (Liblik et al. 2016, see below) include a review of glider-use in scientific research based on investigating 140 peer-reviewed publications and considering the following observing relevant topics:

- Ocean boundary currents

- (Sub)mesoscale processes

- Biogeochemistry

- Biology

- Shallow and marginal seas, coastal areas

- Assimilation, validation, data interpolation, network design

- Noise/ acoustic measurements

- Climate, extension or interpretation of historical data

- Internal waves, tides, turbulence, vertical mixing

- Thermohaline structure, heat content, freshwater content, stratification

- Circulation, current structure

- Sedimentary/geology
These topics include the ones selected by the author of this OS submission.

However, in a second step the Liblik et al. also investigate the integration of data from other observing systems (moorings, Argo, drifter, satellite, . . .) in underwater glider research. For sure the author of this OS submission could not know about the Liblik et al. paper in press.

In summary I am afraid to say that this manuscript has to me no (using OS review criteria 4: poor) scientific significance as no new results are presented. It is a bit strange to judge on scientific quality of a paper rated 4 but here I would say fair (3) - and which is only for the part that investigates the use of underwater glider technology for scientific research by revisiting the published literature. Here my critics are on why exactly this set of publications were selected and other are omitted - more information had to be provided.

The presentation of the manuscript is two to three. The text is written in proper English language, some sentences are not fully appropriate e.g. page 2, line 17: "And the success story of floats should be continued!" – it is unclear to me why this has been written. Is there a risk in float technology to be discontinued? and: What do you mean by "success story of floats"? and finally: why is it relevant in an article on glider technology?

For your information - from my point of view the most relevant papers that provide a comprehensive overview on underwater glider technology and its use in ocean research are:

*(CITED IN MS) Rudnick, DL. 2016. Ocean research enabled by underwater gliders. Annual Review of Marine Science, Vol 8. (Carlson CA, Giovannoni SJ, Eds.).:519-+., Palo Alto: Annual Reviews 10.1146/annurev-marine-122414-033913*

and

*(CITED IN MS) Testor, P. et al. (2010): Gliders as a component of future observing systems, in: Hall, J., Harrison D.E., Stammer, D. (Eds.), Proceedings of the "OceanObs'09: Sustained Ocean Observations and Information for Society" Conference, vol. 2, Venice, Italy, 21-25 September 2009, Vol. 2, ESA Publication WPP-306, 2010.*

and the paper in press in Methods in Oceanography:

*(NOT CITED IN MS) Liblik et al. (in press) Potential for an underwater glider component as part of the Global Ocean Observing System, Methods in Oceanography, http://dx.doi.org/10.1016/j.mio.2016.05.001*

(based on a based on the EU FP7 GROOM project Deliverable D02.01 (NOT CITED IN MS) Current and future GOOS requirements and the need for a glider component (http://www.groom-fp7.eu/lib/exe/fetch.php?media=public: deliverables:groom$_{d20}1_g eomar.pdf$))

---

## Author Comment (AC1) · 5 Jul 2016

Dear anonymous reviewer,

Thank you for your fast response. Please find below my replies to your comments.

1: "All information that is presented has been published elsewhere (most of the papers are cited in the manuscript)."

1: This is true but no other review is bringing together the historical part, the technical part and the scientific part in only one review article and in such a comprehensive way. And as the reviewer stated the author included a paragraph dealing with the "use-of-glider in ocean research, based on publications addressing certain research topics".

2: "I think the title is a bit misleading, although it claims to be "a review" the manuscript is not comprehensive - the author did not use the word "comprehensive" but given the existing, published reviews I would expect that yet another review would be more

comprehensive or is topical e.g. glider use in ocean acoustics – both is not the case."

2: From the authors point of view the title is anything but misleading. The manuscript covers a wide range of topics - from the history and the development of glider technology to its application in a variety of field studies regarding marine sciences. In the manuscript the author clearly pointed out that the manuscript is "a comprehensive review" (Line 17/ Page 3) and actually it is. For instance, Rudnick (2016) cited 146 references. The author cited 183 references. Furthermore, as mentioned before no other review is bringing together the historical part, the technical part AND the scientific part in only one review article and in such a comprehensive way.

3: "In fact it has been mentioned by the author (page 3, line 15) that nothing new is presented in the text and I am somewhat surprised that the manuscript has been sent out for review."

3: Maybe this is a misunderstanding. The author did not write "that nothing new is presented". The author wrote that "This article makes no claim to completeness but is instead intended to describe the main functional systems of these vehicles and to provide a comprehensive review of the studies that have been done to date with the participation of glider technology." Furthermore, in the authors opinion a review article not necessarily needs to present new results.

4: "However, a similar strategy has been published as a report some years ago (GROOM deliverable 2.01, see link below). Moreover, a recent paper in press (Liblik et al. 2016, see below) include a review of glider-use in scientific research based on investigating 140 peer-reviewed publications [. . .]"

4: In the authors opinion there is a difference between a report and a peer-reviewed article with regard to visibility for the scientific community. Moreover, as the reviewer correctly mentioned the report was published some years ago and thus does not include the latest studies. Furthermore, the fact that another review paper (Liblik) is in press highlights the relevance of glider technology and the need to bring more attention to this oceanographic tool. And as the reviewer stated "the author of this OS submission could not know about the Liblik et al. paper in press".

5: "In summary I am afraid to say that this manuscript has to me no (using OS review criteria 4: poor) scientific significance as no new results are presented."

5: In the authors opinion it is not necessary to present new results in a review paper. A review article surveys and summarizes previously published studies, rather than reporting new facts or analysis.

6: "Here my critics are on why exactly this set of publications were selected and other are omitted - more information had to be provided."

6: As mentioned in the table caption the manuscript provides an "Overview of the 5 main application areas of glider technology in ocean sciences and corresponding relevant publications". Relevant in this case means for example visible for the scientific community (citations, availability and so on). Furthermore, this article "makes no claim to completeness but is instead intended to describe the main functional systems of these vehicles and to provide a comprehensive review of the studies that have been done to date with the participation of glider technology". The author is quite willing to include other relevant publications if they correspond to one of the main application areas identified in the article. At this point, the reviewer is encouraged to make suggestions.

7: ""And the success story of floats should be continued!" – it is unclear to me why this has been written. Is there a risk in float technology to be discontinued? and: What do you mean by "success story of floats"? and finally: why is it relevant in an article on glider technology?"

7: This sentence is maybe a bit too emotional but the author intended not only to write a review but also to tell a story. It can be omitted in the next version of the manuscript.

---

## Short Comment (SC1) · 9 Jul 2016

This review is useful for scientists and students starting to work with gliders, whom wish to learn about the origins, as well as the current state of the art. Particularly, Table 1 and the corresponding text are a great catalogue of papers for someone working with glider data in an interdisciplinary way.

However, there are some things I think need to be considered before publication:

1. Although the majority of the paper is well written, there are instances where it would be advantageous to have an English native person read over. For example, on page 1, line 11 – "facilitating a fast access to the world of glider" and on page 2, line 19 – "another technical progress started". Although it is clear what you meant, these words sound a bit strange. Also an 's' should be added to 'glider' when it is plural.

2. Section 1, abbreviations like 'SOFAR' and 'RAFOS' should be explained.

[Figure]

3. abstract, the use of 'drones' in the paper should be avoided by the author as this is associated with warfare and sounds negative.

4. Section 2.2, for Seagliders specifically, the author could mention the different fairing types (standard vs ogive) - Yahnker, Chris. "Overview of the development and advantages of new, larger fairings for the iRobot Seaglider." OCEANS'11 MTS/IEEE KONA. IEEE, 2011. Also concerning section 2.4, ARGOS tags are currently being connected to the tail antenna of Seagliders by a range of working groups as a backup precaution.

5. Section 2.8, lines 20+, the author could perhaps mention time lag (tau) being a problem in some cases (e.g. with oxygen optical sensors). Bittig, Henry C., et al. "Time response of oxygen optodes on profiling platforms and its dependence on flow speed and temperature." Limnology and Oceanography: Methods 12.8 (2014): 617-636.

6. Finally, although the author has mentioned some of problems that could be experienced in the field, perhaps they could be expanded in some instances. A good paper to cite when doing this would be: Queste, Bastien Y., et al. "Deployments in extreme conditions: Pushing the boundaries of Seaglider capabilities." 2012 IEEE/OES Autonomous Underwater Vehicles (AUV). IEEE, 2012.

---

## Referee Comment (RC2) · Anonymous Referee #2 · 20 Sep 2016

I was at first pleased to see the title of this manuscript as i would welcome a thorough, critical review of ocean glider technology. Buoyancy driven gliders are undoubtedly changing the face of oceanography and there is considerable confusion within the ocean observing communities over which platform is best for different purposes and what the true capability of these exciting vehicles is. This manuscript unfortunately only provides a fairly light-touch and quite out of date perspective and so shouldn't be considered a thorough or even accurate review of glider technology. Information provided was lacking in detail throughout the manuscript. For example; there is no discussion of the problems relating to the day-to-day problems glider operators face including thermal inertia issues, calculating glider velocity, glider body compression, oxygen sensor lag, and many other aspects that require expert appraisal. Further, the review is out of date to a quite astounding degree. There are new gliders emerging, both from the recognised and new manufacturers that offer features and capability not even touched upon. The ACSA Seaexplorer and the UW 6000m Deepglider are not

mentioned. I was particularly frustrated that the author recommended one glider over another based on a previous comparison made in a 2001 paper: "It has a finer entry shape than the Slocum electric and therefore is more energy efficient (50% less drag) (Sherman et al., 2001)." This may or may not be the case but more up to date information is required. In summary, this manuscript offers nothing new to inform the Ocean Science readership, and even runs the risk of mis-informing them of the current state of capability of ocean gliders and so should not, in my opinion, be published.
* * *